# Association between Serum 6:2 Chlorinated Polyfluorinated Ether Sulfonate Concentrations and Lung Cancer

**DOI:** 10.3390/toxics12080603

**Published:** 2024-08-19

**Authors:** Weili Mao, Jianli Qu, Ruyue Guo, Yuanchen Chen, Hangbiao Jin, Jingyan Xu

**Affiliations:** 1Department of Pharmacy, Quzhou Affiliated Hospital of Wenzhou Medical University, Quzhou People’s Hospital, Quzhou 324000, China; mwli1987@wmu.edu.cn; 2Key Laboratory of Microbial Technology for Industrial Pollution Control of Zhejiang Province, College of Environment, Zhejiang University of Technology, Hangzhou 310032, China; 1112127008@zjut.edu.cn (J.Q.); gryue1999@163.com (R.G.); hangbiao102@163.com (H.J.)

**Keywords:** 6:2 Cl-PFESA, lung cancer, effect modification, human serum

## Abstract

6:2 chlorinated polyfluorinated ether sulfonate (6:2 Cl-PFESA) exhibits pronounced estrogenic effects, potentially influencing the etiology of lung cancer. This study assessed the potential associations between serum concentrations of 6:2 Cl-PFESA and lung cancer risk at the population level. Odds ratios (ORs) for lung cancer across serum 6:2 Cl-PFESA quartiles were assessed using conditional logistic regression. Additionally, we investigated potential effect modification by various confounding factors. Elevated serum levels of 6:2 Cl-PFESA were consistently associated with an increased risk of lung cancer in both the crude model (OR = 1.62, 95% CI: 1.08–2.42, *p* = 0.018) and the adjusted model (OR = 1.59, 95% CI: 1.06–2.39, *p* = 0.026). Stratified analyses revealed that elevated serum levels of 6:2 Cl-PFESA were associated with increased risk estimates of lung cancer among males (adjusted OR = 2.04, 95% CI: 1.19–3.51, *p* = 0.006), smokers (adjusted OR = 2.48, 95% CI: 1.25–4.89, *p* = 0.003), and drinkers (adjusted OR = 2.20, 95% CI: 0.94–5.16, *p* = 0.049). The results of this study imply that exposure to 6:2 Cl-PFESA at levels considered environmentally relevant may be linked to an elevated risk of developing lung cancer.

## 1. Introduction

Lung cancer is a global health concern, particularly prominent in China, where it stands as the leading cause of cancer-related mortality [1,2]. Globally, lung cancer accounted for the highest incidence (2.1 million cases) and mortality (1.8 million deaths) in 2020, as reported by The World Health Organization [3]. In China, the mortality rate of lung cancer is expected to surge by around 40% from 2015 to 2030, alongside a reported incidence rate of 57.26 per 100,000 population in 2015 [4,5]. Established key risk factors for lung cancer include smoking [6], exposure to air pollution [7] and heavy metal(loid)s [8], and contact with asbestos [9]. Despite the increasing evidence supporting environmental pollutants as potential risk factors for lung cancer, they have garnered comparatively less attention than traditional risk factors [10,11].

An example of such a pollutant is 6:2 chlorinated polyfluorinated ether sulfonate (6:2 Cl-PFESA, commercial name: F-53B). Serving as the primary substitute for perfluorooctane sulfonate (PFOS), 6:2 Cl-PFESA has found extensive application as an anti-mist agent in the Chinese electroplating industry for several decades [12]. The prevailing belief is that dietary intake constitutes the primary pathway through which 6:2 Cl-PFESA enters the human body from the environment. Possessing remarkable biological persistence and accumulation, 6:2 Cl-PFESA exhibits a human half-life of 15.3 years [13], surpassing that of PFOS. The latter two substances were included in the Stockholm Convention for prohibition in 2019 and 2009, respectively [14]. Human serum is considered the primary biomarker for monitoring exposure to 6:2 Cl-PFESA [15]. Research indicates that, apart from PFOS, 6:2 Cl-PFESA is the most consistently detected perfluoroalkyl substance (PFAS) in the serum of the Chinese population [16,17], implying its widespread exposure among the Chinese populace. In addition, epidemiological studies have linked human 6:2 Cl-PFESA exposure to obesity status [18], eye diseases [19], and hypertension in adults [20].

Limited toxicological experiments have explored the impact of 6:2 Cl-PFESA exposure on adverse health outcomes. Experimental evidence demonstrates that 6:2 Cl-PFESA can directly bind to estrogen receptors (ERs) and exert estrogenic effects at doses of 0.5 μM in zebrafish [21]. ERs have been implicated in pivotal roles during the development and progression of non-small-cell lung cancer [22]. Additionally, 6:2 Cl-PFESA exerts a robust binding affinity for peroxisome proliferator-activated receptors (PPARs), as demonstrated through a fluorescence competitive binding assay [23]. Cumulatively, these support the hypothesis positing a potential association between exposure to 6:2 Cl-PFESA and the development of lung cancer. Nevertheless, there is currently no epidemiological evidence examining the potential link between 6:2 Cl-PFESA exposure and lung cancer risk at the population level.

In the present study, we speculate that the general population exposed to 6:2 Cl-PFESA may have an increased risk of developing lung cancer. To validate this hypothesis, we initiated an ongoing prospective cohort known as the Quzhou Environmental Exposure and Human Health (QEEHH) cohort. A nested case-control study was performed within this cohort during the period from 2019 to 2023 in Quzhou, China. This study aimed to investigate the occurrence of 6:2 Cl-PFESA in human serum, to explore the association between concentrations of 6:2 Cl-PFESA in human serum and lung cancer risk, and to examine the potential effect modification by various confounding factors. This investigation encompassed the analysis of exposure characteristics of 6:2 Cl-PFESA in serum and a thorough examination of the association between exposure to this compound and lung cancer risk.

## 2. Methods

Study population. This nested case-control study was performed according to a QEEHH cohort established at the Quzhou People’s Hospital in Quzhou, Zhejiang Province, China. The selection of participants for the case and control groups was conducted within the ongoing prospective QEEHH cohort, specifically designed to investigate environmental factors influencing lung health outcomes. Initially, a total of 39,102 healthy adults were enrolled in the cohort, and a systematic monthly follow-up was conducted. The inclusion criteria for cohort participants were as follows: (1) residency in Quzhou city for more than 5 years and age less than 75 years, and (2) capability to complete their own questionnaire or interview. The criteria for exclusion of patients were as follows: (1) patients with a history of other types of cancer, (2) individuals with chronic diseases that could potentially interfere with the study results (e.g., severe liver or kidney disease), (3) pregnant or breastfeeding women, or (4) patients who refused to provide informed consent. Additionally, individuals from the general population who visited the health management center in the same hospital for physical examinations were extended invitations to partake in the study.

Recruitment commenced in April 2019 and continued for 6 months, resulting in 20,184 participants providing blood samples. Blood samples underwent rapid centrifugation at 4000 rpm for 5 min, and the isolated serum samples were preserved at −80 °C until subjected to experimental analysis. Starting from the initial follow-up until July 2023, participants underwent monthly follow-ups, leading to the confirmation of 329 newly diagnosed lung cancer cases. Diagnosis of all cases of lung cancer was established through radiological examination and tracheoscopy biopsy, with no prior medication or therapies administered. The diagnosed cases belonged to different types of lung cancers, including non-small cell carcinoma (*n* = 168), small cell carcinoma (*n* = 71), and squamous cell carcinoma (*n* = 63). Control recruitment followed a 2:1 matching strategy with cases based on age (±3 years) and sex. Ultimately, the present study included a total of 302 lung cancer cases, and 604 matched healthy individuals as controls. The recruitment flow chart for cases and controls is presented in Appendix A. Each control underwent an additional examination to confirm the absence of any other lung diseases. At the time of enrollment, all subjects provided written informed consent. The research protocol received approval from the Ethics Committee of the Quzhou People’s Hospital.

Serum sample collection and 6:2 Cl-PFESA analysis. All serum samples were collected from the study subjects in April–October 2019. We extracted 6:2 Cl-PFESA from the 906 serum samples using the ion-pair extraction method by Zhao et al. [17] without knowledge of case-control status. Prior to extraction, 200 μL of serum sample was spiked with internal standards (1.5 ng each) and then shaken hand until completely mixed. After that, 4 mL of acetonitrile was added to the serum samples. The mixture was vortexed, sonicated (53 kHz) for 30 min, and centrifuged at 4000 r/min for 10 min. The supernatant was transferred to a new 10 mL PP tube (Biosharp, Beijing, China). The above extraction step was repeated once again with 4 mL of methanol. The eluent was evaporated to near-dryness using gentle nitrogen. The residue was reconstituted in 50 μL of methanol for instrumental analysis.

Quantification of 6:2 Cl-PFESA was performed on an ultraperformance liquid chromatography system (ACQUITY, UPLC) coupled to a tandem mass spectrometer (XEVO_TQS, MS/MS; Waters Co., Milford, MA, USA). A 10 μL aliquot of the sample extract was injected into an Ascentis Express F5 PFP column (2.7 μm, 90 Å, 10 cm × 2.1 mm; Sigma-Aldrich, Oakville, ON, Canada) for chromatographic separation, and the temperature of the column was maintained at 40 °C. The mobile phase consisted of water (containing 0.1% formic acid; A) and methanol (B). The elution gradient started at 20% B, then ramped up to 40% B by 1.0 min, increased to 85% B by 11 min, increased to 100% B by 12 min, and was held at 100% B for 2 min; finally, it was returned to the initial condition. The flow rate of the mobile phase was 0.3 mL/min. The mass spectrometer was operated in the electrospray ionization (negative) and multiple reaction monitoring modes. The detailed descriptions of “standards and reagents”, “sample pre-treatment”, and “parameter of instrument analysis” are shown in the Appendix A.

All labware and solvents were checked to avoid the background 6:2 Cl-PFESA prior to use [24]. Reagent blanks (5 μL of water/methanol, *v*/*v* = 50/50) were injected after each 10 real samples to monitor potential background 6:2 Cl-PFESA carryover [25]. The internal-standard method was applied to quantify the levels of 6:2 Cl-PFESA in human serum. Calibration curves were constructed using six concentration points (0.5, 1, 5, 20, 50, and 100 ng/mL), demonstrating excellent linearity with correlation coefficients exceeding 0.997. Procedural blanks (100 μL of Milli-Q water, *n* = 3) were subjected to identical extraction procedures as the real samples. Procedural blanks showed no detectable presence of 6:2 Cl-PFESA, leading to the determination of the limit of detection (LOD) based on a signal-to-noise ratio of ten. The LOD is presented in Appendix A. Spike recovery experiments conducted using the same extraction method in commercial fetal bovine serum showed the 6:2 Cl-PFESA extraction recoveries ranged from 85 to 113%, with the relative standard deviations (RSD) below 15%. The intra-day and inter-day RSD of 6:2 Cl-PFESA spiked serum samples (*n* = 5) ranged from 2.4 to 9.8% and 5.8 to 13%, respectively (Appendix A).

Data collection. All subjects were instructed to complete a standardized questionnaire during face-to-face interviews conducted by highly trained nurses. Various sociodemographic and lifestyle characteristics were gathered, including but not limited to sex, age, educational level, height, weight, annual household income, smoking habit, alcohol consumption habit, family history of lung cancer, and history of any lung disease. Current smokers were defined as individuals who reported smoking at least one cigarette per day for a duration exceeding one year [26]. Participants reporting regular alcohol consumption during the interview were categorized as current drinkers. The ratio of the subject’s weight to the square of their height (in kilograms per meter squared) was used to obtain the body mass index (BMI). Individuals with a history of lung disease were identified as those who had been previously diagnosed with bronchitis, chronic obstructive pulmonary disease, emphysema, or pneumonotomy. The lifestyle and dietary habits of cases remained relatively stable from the time of blood collection to the diagnosis of lung cancer, as assessed through monthly follow-ups. Similarly, selected controls exhibited minimal changes in lifestyle and dietary habits compared to the period preceding serum collection.

Statistical analysis. To normalize the data between the control and lung cancer groups, we used propensity score matching to ensure comparable baseline characteristics, such as age, gender, and smoking status. We also applied log transformation to the serum 6:2 Cl-PFESA concentrations to normalize the distribution and reduce skewness. The distributions of 6:2 Cl-PFESA concentrations were assessed through the Shapiro–Wilk test. Due to the left-skewed nature of 6:2 Cl-PFESA concentrations, we utilized the Mann–Whitney U test to compare the serum concentrations of 6:2 Cl-PFESA between the case and control groups. Sociodemographic and lifestyle characteristics differences between cases and controls were assessed using the chi-squared test. Associations of serum 6:2 Cl-PFESA concentrations with lung cancer risk were examined using conditional logistic regression models. This involved calculating matched odds ratios (ORs) along with their 95% confidence intervals (CIs). To fit the models, serum 6:2 Cl-PFESA concentrations were treated as categorical variables. 6:2 Cl-PFESA concentrations were stratified into quartiles according to their distribution in the control group, with the first quartile designated as the reference group. Linear trends were assessed by treating the median values of 6:2 Cl-PFESA quartiles as a continuous variable. The statistical significance of this p-trend value was determined using the Wald test.

In this investigation, potential confounding factors were *a priori* selected based on directed acyclic graphs (Appendix A). The covariates encompassed sociodemographic factors [age (continuous), sex (male, female), BMI (<18.5, 18.5–24.9, >24.9; kg/m^2^), annual household income (<50,000, 50,000–100,000, 100,000–150,000, >150,000; CNY), educational level (high school or lower, junior college or undergraduate, postgraduate or above)] and lifestyle characteristics [smoking habit (nonsmoker, current smoker), alcohol consumption habit (nondrinker, current drinker)] within the final models.

Sensitivity analyses were conducted, excluding subjects with a family history of lung cancer (yes, no) and those ever diagnosed with any lung cancer (yes, no). In sensitivity analyses, if any member of the case or either of the two matched controls met these criteria, the entire matched set was censored. Sex-stratified analyses were employed to explore potential effect modification by the sex of subjects. A multiplicative interaction term between 6:2 Cl-PFESA and the sex of subjects was introduced to evaluate the interaction effect in the models, with the 6:2 Cl-PFESA concentrations categorized into quartiles. Additionally, we explored potential effect modifications of smoking habits on the association between 6:2 Cl-PFESA and lung cancer because previous studies have indicated that smoking is a crucial risk factor influencing the development of lung cancer. In these stratification analyses, a cross-product term of 6:2 Cl-PFESA and smoking habit was included. Statistical analyses were performed using SAS 9.4 (SAS Institute Inc., New York, NY, USA), and statistical significance was set at two-sided *p* < 0.05.

## 3. Results

The concentrations of 6:2 Cl-PFESA in 302 cases and 604 control pairs under various demographic parameters are presented in Table 1. The mean age of cases was 62.1 ± 13.7 years, and for controls, it was 62.8 ± 13.8 years, with a mean BMI of 23.2 ± 3.5 kg/m^2^ for cases and 23.4 ± 3.4 kg/m^2^ for controls. No significant differences were observed between cases and controls for BMI, education level, annual household incomes, occupational status, marital status, dietary habit, residence, smoking habit, and drinking habit (all *p* > 0.05). In comparison to the controls, cases showed a notably higher prevalence of individuals with a family history of lung cancer (*p* < 0.004) and a prior history of any lung disease (*p* < 0.003).

The detection rates for 6:2 Cl-PFESA in the serum were 98.7% for both cases and controls (Appendix A). In the cases, the median serum concentration of 6:2 Cl-PFESA was 3.2 ng/mL with an interquartile range of 1.5–6.2 ng/mL, while in the control subjects, it was 2.9 ng/mL with an interquartile range of 1.3–6.0 ng/mL (Figure 1). Median concentrations of 6:2 Cl-PFESA did not significantly differ between cases and controls (Mann–Whitney *U* test, *p* > 0.05).

Table 2 illustrates the relationship of serum 6:2 Cl-PFESA concentrations across quartiles with lung cancer risk. In comparison to the first quartile of serum 6:2 Cl-PFESA concentrations, a notable elevation in lung cancer risk was evident in the crude model (*p* for trend = 0.018). Following adjustment for multiple covariates (including gender, age, BMI, education level, annual household incomes, occupational status, marital status, dietary habit, residence, smoking habit, and alcohol consumption habit), a comparable and marginally significant trend persisted in the association between lung cancer risk and serum 6:2 Cl-PFESA concentrations (*p* for trend = 0.026). An analysis excluding subjects with a family history of lung cancer found that the significant association of 6:2 Cl-PFESA with lung cancer remained essentially unchanged (OR = 1.73, 95% CI: 1.13, 2.65 for the top vs. bottom quartile). Excluding 23 controls and 26 cases with a history of any lung cancer, the adjusted ORs for the second, third, and highest quartiles were 1.32 (95% CI: 0.84, 2.05), 1.59 (95% CI: 1.03, 2.45), and 1.73 (95% CI: 1.13, 2.65), respectively, indicating no substantial change. Excluding 683 urban participants rendered the association between 6:2 Cl-PFESA and lung cancer risk statistically insignificant.

We conducted a stratified analysis to investigate variations in the associations of 6:2 Cl-PFESA with lung cancer according to the subjects’ sex (Table 3). Among males, the risk of lung cancer demonstrated a significant increase with rising quartiles of 6:2 Cl-PFESA (*p* for trend = 0.006); however, no significant association was detected between 6:2 Cl-PFESA and lung cancer risk among females (*p* for trend = 0.840). No significant interaction effect was observed between serum 6:2 Cl-PFESA concentrations and subjects’ sex concerning lung cancer risk (*p* for interaction = 0.232).

Additionally, we investigated the potential modification effect of smoking habits on the associations of 6:2 Cl-PFESA with lung cancer (Table 3). Among smokers, the risk of lung cancer demonstrated a significant increase with rising quartiles of 6:2 Cl-PFESA, yielding an adjusted OR of 2.48 (95% CI: 1.25, 4.89) for the top vs. bottom quartile (*p* for trend = 0.003). Conversely, among nonsmokers, no clear linear increasing trend was observed between 6:2 Cl-PFESA and lung cancer risk (*p* for trend = 0.662). The interaction between 6:2 Cl-PFESA and smoking habit did not reach statistical significance (*p* for interaction = 0.216). After adjusting for confounder factors, the association of serum 6:2 Cl-PFESA with lung cancer risk was more pronounced in drinkers (*p* for trend = 0.049) than in non-drinkers (Appendix A).

## 4. Discussion

Detection of 6:2 Cl-PFESA was observed in over 98% of serum samples, suggesting ubiquitous exposure of the study population to 6:2 Cl-PFESA. Several studies have documented concentrations of 6:2 Cl-PFESA in the serum of the general Chinese population. In 2022, Zhao et al. [17] reported high detection rates and levels of 6:2 Cl-PFESA (detection rates 100%, median 1.2 ng/mL) in the serum of the general Hangzhou population, consistent with our findings in controls. Jin et al. [27] reported that our study’s general population had higher serum levels of 6:2 Cl-PFESA (median 2.9 ng/mL) than populations in Beijing (1.4 ng/mL), Shijiazhuang (1.59 ng/mL), Jinan (2.53 ng/mL), and Shenzhen (1.57 ng/mL), but lower than those in Tianjin (3.83 ng/mL), Shouguang (4.87 ng/mL), and Wuhan (3.01 ng/mL). Compared to other studies on the general population, our study revealed lower serum levels of 6:2 Cl-PFESA than those reported in Wuhan (median 4.78 ng/mL) [13], and Huantai (4.04 ng/mL) [28], yet higher concentrations than those observed in Shenyang [18] and Shijiazhuang (median 1.79 ng/mL) [29]. In light of the prevalent exposure to 6:2 Cl-PFESA among Chinese populations, there is a need for increased attention to its toxicity and epidemiological effects in the general population.

This study demonstrates a significant association between elevated human serum 6:2 Cl-PFESA concentrations and an increased risk of lung cancer. To date, no study has investigated the potential association between human serum concentrations of 6:2 Cl-PFESA and the risk of lung cancer. Previous epidemiological studies have suggested associations between women’s exposure to 6:2 Cl-PFESA and polycystic ovarian syndrome in infertile women, gestational diabetes mellitus in pregnant women, and altered glucocorticoid levels in newborns [30,31,32]. In a birth cohort of 374 Chinese neonates, exposure to serum 6:2 Cl-PFESA was found to correlate with elevated 11-deoxycortisol in boys [OR = 8.38 (95% CI: 1.16, 16.11) increase for a one-unit increase in the ln-transformed 6:2 Cl-PFESA level], and cortisol [13.13 (95% CI: 4.47, 22.52)] in girls [30]. This evidence suggests a potential endocrine-disrupting effect of 6:2 Cl-PFESA. The statistical significance of these observed associations remained unchanged when we excluded subjects with a family history of lung cancer or any history of lung disease, both of which are recognized as robust risk factors for lung cancer.

One plausible mechanism by which 6:2 Cl-PFESA may contribute to lung cancer is through the activation of ER pathways, thus facilitating the initiation and progression of the disease. An in vivo study revealed that exposure to 6:2 Cl-PFESA influenced the expression of testosterone, androstenedione, progesterone, cortisol, corticosterone, and 11-deoxycorticosterone in rat serum [33]. Similarly, another study in zebrafish zygotes displayed a notable increase in the expression levels of ERα, ERβ1, ERβ2, and CYP19a after exposure to 6:2 Cl-PFESA [21]. ERs have been established as playing a crucial role in maintaining lung tissue function [22]. Consequently, exposure to 6:2 Cl-PFESA may impact lung function due to its robust affinity for ERs, potentially contributing to the development of non-small-cell lung cancer.

In the stratified analysis, a notable observation was the significant association of serum 6:2 Cl-PFESA concentrations with lung cancer risk among smokers and drinkers, even though the effect modification was not pronounced. Smoking is widely acknowledged as a potent risk factor for the onset and progression of lung cancer, where individuals who smoke have a higher baseline risk of lung cancer than those who do not smoke [34,35,36]. Alcohol is also a well-known risk factor for lung cancer. Acetaldehyde, the metabolite of alcohol, may induce cancer by disrupting DNA function [37]. In stratification by sex, males exhibited a heightened risk of lung cancer with increasing serum 6:2 Cl-PFESA concentrations, albeit with the *p*-value for the interaction effect exceeding 0.05. One plausible explanation for sex-specific results could be the higher prevalence of smoking and drinking habits among male participants. This could be partly attributed to the elevated prevalence of smoking and drinking habits, both recognized risk factors for lung cancer, in male participants. These factors may potentially exert a synergistic effect on the association of BTH and 2-OH-BTH with the development of lung cancer.

The nested case-control study adhered to rigorous protocols and guidelines, ensuring the robustness and reliability of our findings—an essential strength of our investigation. For instance, we systematically accounted for a range of potential confounding variables related to lung cancer. In addition, there was minimal alteration in lifestyle and dietary habits among cases from serum collection to lung cancer diagnosis, which implies little change in exposure to 6:2 Cl-PFESA in the short term. However, this study has several limitations. First of all, owing to the presence of mixed chemicals in the environment, co-exposure among 6:2 Cl-PFESA and other environmental contaminants (such as other PFASs, parabens, and phthalates) may have confounding effects on lung cancer risk. Then, air pollution was recognized as a significant risk factor for lung cancer [38,39]. Despite this study performing a sensitivity analysis for both rural and urban populations, evaluating the impact of air pollution on the association of 6:2 Cl-PFESA with lung cancer risk in urban residence people was challenging due to insufficient data on air pollution. In addition, one of the big limitations of this study is that human exposure to heavy metals (through air and contaminated fruits and vegetables) was not considered a risk factor for the development of lung cancer. 

## 5. Conclusions

Utilizing a nested case-control design involving 906 Chinese participants, this study revealed a significant correlation between elevated serum levels of 6:2 Cl-PFESA and an increased risk of lung cancer for the first time. Stratified analyses further revealed that elevated serum levels of 6:2 Cl-PFESA were associated with increased risk estimates of lung cancer among males, and drinkers. The strengths of our study included comprehensive data collection, which encompasses detailed dietary information, environmental exposure data, and health records, allowing for a nuanced analysis of the associations between human serum 6:2 Cl-PFESA concentrations and lung cancer risk. The study’s cultural relevance, particularly the consideration of dietary habits like spicy versus non-spicy foods, enhances the accuracy and applicability of our findings within the Chinese population. Additionally, the consideration of potential confounding factors further strengthens the validity of our results. The results of this study imply that exposure to 6:2 Cl-PFESA at levels considered environmentally relevant may be linked to an elevated risk of lung cancer. Further mechanistic studies are necessary to elucidate the dose-dependent associations between 6:2 Cl-PFESA exposure and the risk of lung cancer.

## Figures and Tables

**Figure 1 toxics-12-00603-f001:**
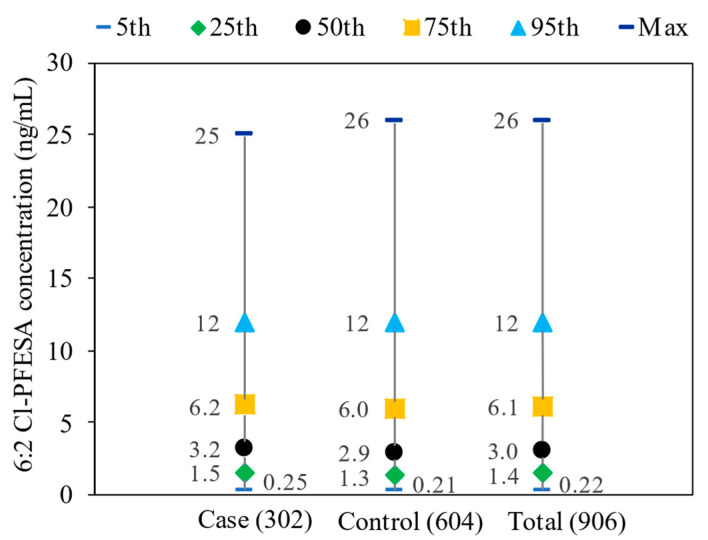
Distributions of serum 6:2 Cl-PFESA concentration (ng/mL) among cases (*n* = 302), controls (*n* = 604), and total participants (*n* = 906).

**Table 1 toxics-12-00603-t001:** Serum concentrations (ng/mL) of 6:2 Cl-PFESA among participants with different characteristics (*n* = 906).

Characteristics	Control (*n* = 604)				Case (*n* = 302)				*p*-Value *^a^*
	Number (%)	Mean	Median	Range	Number (%)	Mean	Median	Range	
**Gender**									N/A
Male	356 (58.9)	4.0	2.9	<LOD-26	178 (58.9)	4.6	3.3	<LOD-25	
Female	248 (41.1)	4.0	2.9	<LOD-25	124 (41.1)	3.7	3.1	<LOD-13	
**Age (years)**									N/A
<51	110 (18.2)	3.7	2.8	<LOD-26	55 (18.2)	4.8	3.6	<LOD-25	
51–60	120 (19.9)	4.2	3.0	0.031–21	60 (19.9)	4.4	3.4	0.11–12	
61–70	200 (33.1)	4.3	2.9	<LOD-25	100 (33.1)	3.9	3.2	0.10–15	
>70	174 (28.8)	3.7	2.9	<LOD-20	87 (28.8)	4.1	3.1	<LOD-20	
**Body mass index (kg/m^2^)**									0.842
<18.5	38 (6.3)	3.5	2.8	0.033–20	22 (7.3)	3.6	3.1	<LOD-13	
18.5–24.9	375 (62.1)	4.0	2.9	<LOD-26	187 (61.9)	4.4	3.3	<LOD-20	
>24.9	191 (31.6)	4.0	2.8	<LOD-25	93 (30.8)	4.0	3.2	<LOD-25	
**Educational level**									0.681
≤9	494 (81.8)	4.1	2.9	<LOD-25	242 (80.1)	4.2	3.2	<LOD-22	
>9	108 (17.9)	3.8	2.8	<LOD-26	57 (18.9)	4.1	3.4	<LOD-25	
**Annual household income (CNY)**									0.626
<50,000	216 (35.8)	3.8	2.8	<LOD-25	113 (37.4)	4.1	3.2	<LOD-15	
50,000–100,000	248 (41.0)	4.1	2.9	<LOD-26	114 (37.8)	4.5	3.2	<LOD-25	
>100,000	140 (23.2)	4.2	3.1	<LOD-23	75 (24.8)	4.0	3.6	<LOD-22	
**Occupational status**									0.330
Employed	213 (35.3)	4.1	2.9	<LOD-23	97 (32.1)	4.9	3.3	0.10–25	
Unemployed	389 (64.4)	4.0	2.9	<LOD-26	205 (67.9)	3.9	3.2	<LOD-14	
**Marital status**									0.542
Married	498 (82.5)	4.0	2.8	<LOD-25	244 (80.8)	4.2	3.3	<LOD-25	
Divorced and widowed	106 (17.5)	3.9	3.1	<LOD-26	58 (19.2)	4.0	3.1	<LOD-22	
**Dietary habit**									0.909
Spicy food	474 (78.5)	4.2	2.9	<LOD-26	236 (78.1)	4.2	3.2	<LOD-25	
Non-spicy food	130 (21.5)	3.4	2.9	<LOD-20	66 (21.9)	4.2	3.7	0.10–14	
**Residence**									0.957
Urban	455 (75.3)	3.9	2.8	<LOD-26	228 (75.5)	4.2	3.2	<LOD-25	
Rural	149 (24.7)	4.3	3.1	<LOD-23	74 (24.5)	4.1	3.2	0.10–22	
**Smoking habit**									0.565
Nonsmoker	368 (61.0)	4.0	2.9	<LOD-25	178 (58.9)	4.0	3.1	<LOD-25	
Current smoker	236 (39.0)	4.1	2.8	<LOD-26	124 (41.1)	4.6	3.4	<LOD-20	
**Alcohol consumption habit**									0.304
Nondrinker	418 (69.2)	3.9	2.9	<LOD-25	219 (72.5)	4.1	3.2	<LOD-22	
Current drinker	186 (30.8)	4.2	2.9	<LOD-26	83 (27.5)	4.6	3.2	<LOD-25	
**Family history of lung cancer**									0.004
No	583 (96.5)	4.0	2.9	<LOD-26	279 (92.4)	4.3	3.2	<LOD-25	
Yes	16 (2.7)	3.7	2.5	0.29–10	20 (6.6)	3.9	4.1	0.54–8.6	
**History of any lung disease *^b^***									0.003
No	580 (96.0)	4.0	2.9	<LOD-26	275 (91.1)	4.3	3.3	<LOD-25	
Yes	23 (3.9)	3.1	2.3	<LOD-13	26 (8.6)	3.3	2.8	0.11–10	
**Histologic type**									N/A
Non-small cell carcinoma	N/A	N/A	N/A	N/A	168 (55.6)	4.3	3.2	<LOD-20	
Small cell carcinoma	N/A	N/A	N/A	N/A	71 (23.5)	3.9	3.5	0.11–11	
Squamous cell carcinoma	N/A	N/A	N/A	N/A	63 (20.9)	4.1	2.9	<LOD-25	

Note: N/A, not applicable. *^a^ p*-Values were calculated to examine the demographic distribution between cases and controls using the chi-square test. *^b^* lung disease included participants who had previously been diagnosed with bronchitis, chronic obstructive pulmonary disease, emphysema, and pneumonedema.

**Table 2 toxics-12-00603-t002:** Association between serum concentrations of 6:2 Cl-PFESA and lung cancer risk.

	Cases/Controls (*n*)	Crude	Adjusted *^a^*
		OR (95% CI)	OR (95% CI)
Total participants (*n* = 906)			
Q1 (<1.3)	56/151	Ref.	Ref.
Q2 (1.3–2.9)	68/151	1.21 (0.79, 1.84)	1.20 (0.79, 1.85)
Q3 (2.9–6.0)	87/151	1.55 (1.03, 2.32)	1.51 (1.00, 2.28)
Q4 (>6.0)	91/151	1.62 (1.08, 2.42)	1.59 (1.06, 2.39)
*p* for trend *		0.018	0.026
Excluding participants who have a family history of lung cancer (*n* = 774)			
Q1 (<1.3)	46/129	Ref.	Ref.
Q2 (1.3–2.8)	58/129	1.26 (0.79, 1.99)	1.23 (0.77, 1.96)
Q3 (2.8–5.7)	77/129	1.67 (1.07, 2.59)	1.60 (1.02, 2.50)
Q4 (>5.7)	77/129	1.67 (1.07, 2.59)	1.63 (1.04, 2.55)
*p* for trend *		0.027	0.036
Excluding participants who were diagnosed with any lung disease (*n* = 753)			
Q1 (<1.3)	46/126	Ref.	Ref.
Q2 (1.3–2.9)	58/125	1.27 (0.80, 2.01)	1.20 (0.75, 1.95)
Q3 (2.9–6.0)	71/126	1.54 (0.98, 2.41)	1.42 (0.89, 2.25)
Q4 (>6.0)	76/125	1.66 (1.07, 2.59)	1.62 (1.02, 2.55)
*p* for trend *		0.028	0.037
Excluding urban participants (*n* = 223)			
Q1 (<1.6)	14/37	Ref.	Ref.
Q2 (1.6–3.1)	19/38	1.32 (0.57, 3.01)	1.35 (0.55, 3.31)
Q3 (3.1–5.3)	20/37	1.42 (0.63, 3.20)	1.38 (0.57, 3.32)
Q4 (>5.3)	21/37	1.51 (0.66, 3.44)	1.55 (0.63, 3.77)
*p* for trend *		0.392	0.375

Note: OR, odds ratio; CI, confidence interval; Q, quartiles; Ref, reference. *^a^* Adjusted by sex, age, BMI, education level, annual household income, smoking habit, and alcohol consumption history. * *p*-Value was defined as FDR-corrected *p* < 0.05.

**Table 3 toxics-12-00603-t003:** Association between 6:2 Cl-PFESA and lung cancer risk stratified by subjects’ gender and smoking habit.

	Cases/Controls (*n*)	Crude	Adjusted *^a^*
		OR (95% CI)	OR (95% CI)
Subjects’ gender			
Male (*n* = 534)			
Q1 (<1.3)	30/89	Ref.	Ref.
Q2 (1.3–2.9)	38/89	1.26 (0.72, 2.22)	1.27 (0.71, 2.27)
Q3 (2.9–5.9)	49/89	1.63 (0.95, 2.80)	1.64 (0.94, 2.87)
Q4 (>5.9)	61/89	2.03 (1.20, 3.44)	2.04 (1.19, 3.51)
*p* for trend *		0.005	0.006
Female (*n* = 372)			
Q1 (<1.4)	28/62	Ref.	Ref.
Q2 (1.4–2.9)	28/62	1.00 (0.53, 1.88)	0.99 (0.50, 1.94)
Q3 (2.9–6.1)	41/62	1.46 (0.80, 2.65)	1.43 (0.75, 2.72)
Q4 (>6.1)	27/62	0.96 (0.51, 1.82)	1.04 (0.53, 2.03)
*p* for trend *		0.970	0.840
*p* for interaction		0.239	0.232
Smoking habit			
Smoker (*n* = 360)			
Q1 (<1.3)	21/59	Ref.	Ref.
Q2 (1.3–2.8)	22/59	1.04 (0.52, 2.10)	1.10 (0.51, 2.34)
Q3 (2.8–5.7)	36/59	1.71 (0.89, 3.27)	1.86 (0.92, 3.75)
Q4 (>5.7)	45/59	2.14 (1.14, 4.02)	2.48 (1.25, 4.89)
*p* for trend *		0.007	0.003
Nonsmoker (*n* = 546)			
Q1 (<1.4)	36/92	Ref.	Ref.
Q2 (1.4–2.9)	47/92	1.30 (0.77, 2.19)	1.32 (0.77, 2.25)
Q3 (2.9–6.1)	51/92	1.41 (0.84, 2.37)	1.38 (0.81, 2.35)
Q4 (>6.1)	44/92	1.22 (0.72, 2.07)	1.22 (0.71, 2.09)
*p* for trend *		0.641	0.662
*p* for interaction		0.244	0.216

Note: CI, confidence interval; Q, quartiles; Ref, reference. *^a^* Adjusted by sex, age, BMI, education level, annual household incomes, smoking history, and alcohol consumption history. * *p*-Value was defined as FDR-corrected *p* < 0.05.

## Data Availability

Data set available on request from the authors.

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
