# Peer review of "Association between Serum 6:2 Chlorinated Polyfluorinated Ether Sulfonate Concentrations and Lung Cancer"

_toxics, 2024, doi:10.3390/toxics12080603_

Round 1

Reviewer 1 Report

Comments and Suggestions for Authors

In general, the health risks of 6:2 Cl-PFESA in the population are still unknown. The conducted research in general could help to understand the role of chlorinated polyfluorinated ether sulfonate in adverse outcomes. However, the manuscript has to be improved because some parts of the manuscript lack information that should be added. The major issue is that the classification of lung cancer is missing based on the pathohistological analysis.

Introduction section, line 40-42: these factors need to be expanded. See recently published paper in this journal regarding lung cancer for more details (https://www.mdpi.com/2305-6304/12/7/490).

Introduction section, line 55-56: references are missing

Introduction section, line 58: add more epidemiological studies

Introduction section, line 63-64: non-small cell lung cancer (adenocarcinoma) is depended on hormonal status and estrogen receptors might impact the progression of non-small-cell lung cancer not all type of lung cancer. This comment should also be taken into the consideration in the discussion part (283-285)

Introduction section, line 67-68: multiple studies have demonstrated that PPARγ activation exerts anti-tumor effects in lung cancer. The role of ppar receptors is contradictory so please rewrite this sentence in order to be more appropriate. This comment should also be taken into the consideration in the discussion part (287-295)

Introduction section 60-71: In vitro studies conducted on cell lines are missing, hence this paragraph needs to be rewritten

Introduction section, line 73-74: Sentence should be rewritten.

Introduction section, line 80-81:  Sentence should be deleted.

Study population, inclusion and exclusion criteria should be expanded. Why only 604 were used as control? These criteria need to be well documented in order to avoid any misunderstanding and misinterpretation.

Line 111: Why 908 serums? 302+604=906

Dietary habit: please justify the relevance of spicy and not spicy food

Having in mind that median concentrations did not significantly differ between cases and controls authors could not conclude that 6:2 Cl-PFESA is a risk factor for lung cancer. Please be more careful in choosing proper words in discussion and conclusion as well as in abstract section.

Line 261-263: sentence has to be rewritten

Line 275-277: delete

Conclusion should be based on the obtained results not to be speculative

Comments on the Quality of English Language

minor improvements are needed

Reviewer 2 Report

Comments and Suggestions for Authors

Between 2019 and 2023, 906 participants were recruited for this study, with 302 lung cancer patients and 604 controls providing serum samples. They looked at the potential link between 6:2-Cl-PFESA serum concentrations and lung cancer risk at the community level. Overall, the study is well-designed, interesting and the findings are clearly presented. The findings shed new light on the potential effects of 6:2 Cl-PFESA on lung cancer risk and could make a valuable addition to the field of toxicology.  

1) Was there any correlation found between the serum concentrations of 6:2 Cl-PFESA and the lung biomarkers, CYFRA 21-1, SCCA, and CEA?

2) How were the serum concentrations of 6:2 Cl-PFESA measured by the authors? Could you please describe the Materials & Methods procedure?

3) How was the quality of each serum sample assessed? What was the acceptable range for each sample?

4) How was the data normalized between control & lung cancer group.

5) Can you provide a graphical abstract?

Reviewer 3 Report

Comments and Suggestions for Authors

In the manuscript titled “Association between Serum 6:2 Chlorinated Polyfluorinated Ether Sulfonate Concentrations and Lung Cancer.”, the authors investigates the association between serum concentrations of 6:2 chlorinated polyfluorinated ether sulfonate (6:2 Cl-PFESA) and the risk of lung cancer. Conducted as a nested case-control study in Zhejiang Province, China, from 2019 to 2023, it examines 302 incident lung cancer cases confirmed up to July 2023. The study measures serum concentrations of 6:2 Cl-PFESA at cohort entry and uses conditional logistic regression to assess the odds ratios (ORs) for lung cancer across different serum concentration quartiles. Additionally, the study explores the potential modification effects of various confounding factors such as gender, socioeconomic status, smoking status, drinking habits, and dietary habits.

This study represents the first investigation into the potential association between exposure to 6:2 Cl-PFESA and lung cancer risk, suggesting that exposure to environmentally relevant levels of 6:2 Cl-PFESA may increase the risk of developing lung cancer. Moreover, although the study has limitations, which the authors thoroughly address in the text, the results are of great importance to the scientific community and underscore the need for further research to elucidate the dose-dependent relationships and underlying mechanisms of this association.

From a lexical and formal perspective, the paper is well-constructed. The authors employ clear and precise language to convey their findings, making the content accessible to readers with relevant scientific backgrounds. Furthermore, the manuscript features well-organized tables and figures that effectively summarize the data and complement the textual content. The use of citations and references is consistent and appropriate, providing a solid foundation for the study’s assertions and placing it within the broader context of existing research. Therefore, I have only a few suggestions to make to improve the manuscript for publication in Toxics. For these reasons, I decided to accept the manuscript after minor revisions

Minor points

Abstract Section: According to the Guidelines for authors, the abstract should be a total of about 200 words maximum. Therefore, I suggest to reduce the number of words; 

The Introduction Section is well structured and clear. However, the authors should describe in more detail what the objectives of the study were.

The formatting in Table 1 needs to be improved.

The conclusions section needs to be completely revised. The authors should highlight the strengths of the study to underscore its findings, particularly given that it is the first investigation into the potential link between exposure to 6:2 Cl-PFESA and lung cancer risk.

References Section: This section should be revised according to the guidelines for authors. Furthermore, the authors need to adapt the bibliography cited in the text in accordance with the journal’s guidelines.

Round 2

Reviewer 1 Report

Comments and Suggestions for Authors

Authors try to investigate the possible linkage between serum 6:2 Cl-PFESA concentration in Chinese population and the incidence of lung cancer. Although authors have made improvements and this is a topic of interest, there are some concerns that cannot be ignored, especially regarding the methodology approach.

Based on the revised lines presented in the manuscript all diagnosed cases belonged to the non-small type of lung cancer (line 134). However, based on the presented results in Table 1 (Serum concentrations (ng/mL) of 6:2 Cl-PFESA among participants with different characteristics (n = 906)), the diagnosed cases belonged to different types of lung cancers. Particularly Non-small cell carcinoma (n = 168), small cell carcinoma  (n= 71) and squamous cell carcinoma (n=63).

Moreover, the criteria for inclusion and exclusion of patients into the study are not clear.

Comments on the Quality of English Language

The quality of english language is appropriate

Reviewer 2 Report

Comments and Suggestions for Authors

The authors have included all suggestions into the updated manuscript. I don't have any additional comments. 

Author Response

Thanks for your comments.

Round 3

Reviewer 1 Report

Comments and Suggestions for Authors

The authors made improvements.

The part related to the limitations of study should be expanded having in mind that exposure to heavy metals through the air as well as food is common in China. 

 The revised manuscript contains part related to the strenght and limitations of the performed research (given in lines 442-460).

However, one of the big limitations of this study is that exposure to heavy metals were not considered as risk factor for the development of lung cancer (line 453). Heavy metal pollution is a global problem in China, hence,  air exposure to heavy metals as well as exposure to heavy metals though contaminated fruits and vegetables increases the risk of lung cancer development and has a negative impact on the overall survival of lung cancer patients based on the epidemiological studies. Bearing in mind all above mentioned, authors  should include sentence related to heavy metals as one of the limitations of this research in the revised manuscript.  
